# Piezoresistance Characterization of Silicon Nanowires in Uniaxial and Isostatic Pressure Variation

**DOI:** 10.3390/s22176340

**Published:** 2022-08-23

**Authors:** Elham Fakhri, Rodica Plugaru, Muhammad Taha Sultan, Thorsteinn Hanning Kristinsson, Hákon Örn Árnason, Neculai Plugaru, Andrei Manolescu, Snorri Ingvarsson, Halldor Gudfinnur Svavarsson

**Affiliations:** 1Department of Engineering, Reykjavik University, Menntavegur 1, 102 Reykjavik, Iceland; 2National Institute for Research and Development in Microtechnologies-IMT Bucharest, 077190 Voluntari, Romania; 3Science Institute, University of Iceland, Dunhaga 3, 107 Reykjavik, Iceland

**Keywords:** silicon nanowires, MACE, piezoresistivity

## Abstract

Silicon nanowires (SiNWs) are known to exhibit a large piezoresistance (PZR) effect, making them suitable for various sensing applications. Here, we report the results of a PZR investigation on randomly distributed and interconnected vertical silicon nanowire arrays as a pressure sensor. The samples were produced from p-type (100) Si wafers using a silver catalyzed top-down etching process. The piezoresistance response of these SiNW arrays was analyzed by measuring their I-V characteristics under applied uniaxial as well as isostatic pressure. The interconnected SiNWs exhibit increased mechanical stability in comparison with separated or periodic nanowires. The repeatability of the fabrication process and statistical distribution of measurements were also tested on several samples from different batches. A sensing resolution down to roughly 1m pressure was observed with uniaxial force application, and more than two orders of magnitude resistance variation were determined for isostatic pressure below atmospheric pressure.

## 1. Introduction

Low-dimensional structures may possess unique mechanical, electrical, optical, and thermoelectric properties. Particularly, silicon nanowires (SiNWs) have demonstrated properties suitable for various advanced applications [1,2,3], including low-cost thermoelectric devices and chemo-biological sensors with ultrahigh sensitivity [4,5]. The integration of SiNWs in electronic devices is favoured by their compatibility with the well-established Si-SiO_2_ electronic industrial technology. Bulk silicon has been known for a while to exhibit high piezo resistance (PZR) effect [6]. In bulk semiconductors, the PZR-effect takes place, in principle, due to a change in the electronic structure and modification of the charge-carriers effective masses. This phenomenon has found practical applications in many Si-based devices, such as pressure transducers [7], cantilevers for atomic force microscopy [8], accelerometers [9], biosensors [10], and multi-axis force sensing tools [11].

Recently, nanowires have been shown to possess the ability to significantly increase the PZR response [12]. A giant PZR was observed in p-doped SiNWs with diameters of 50 nm–350 nm and a length of microns initially under tensile uniaxial stress [13]. However, the PZR effect in n-doped nanowires was found to be comparable to that in the bulk counterpart, both for tensile and compressive uniaxial stress [14].

On the theoretical side, the origin of the PZR effect in SiNWs has long been under debate, and most frequently, it is referred to as anomalous PZR [15]. It has been related to quantum confinement effects [16], surface charge effects [17,18,19], strain-induced bandgap shift [20], or changes in the charge carrier’s effective masses [21]. A complex model incorporating these mechanisms has been proposed in order to analytically quantify the PZR effect in silicon [22].

A survey of several PZR sensors based on SiNWs, e.g., cantilever [23], opto-mechanical sensor [24], flexible pressure sensor [18], or breath detector [19], shows that different methods have been used for fabricating the SiNWs, such as vapor-liquid-solid (VLS), laser ablation, and metal-assisted catalyzed etching (MACE) [25]. Among these methods, MACE is the simplest and most versatile one [26]. It relies on catalyzed etching with assistance from a perforated metal template film (typically gold or silver) [27] or randomly distributed metallic nanoparticles (typically gold or silver) [28,29] spread on the Si-wafer. To date, studies have been focused on using different gas types to apply direct pressure on SiNWs, either single SiNW or arrays of SiNWs [19,30], and have neglected the SiNWs response under isostatic pressure, which creates a load uniformly distributed on the sample surface. Here, we report on the PZR effect in SiNWs obtained by MACE under uniaxial compression load as well as isostatic pressure in a vacuum chamber. We find that the interconnected SiNWs are mechanically stronger and functionally more stable compared with the arrays of separated wires under applied uniaxial pressure. They show higher PZR sensitivity under isostatic pressure variation. We also demonstrate a simple, low-cost, and reproducible fabrication method for a robust and sensitive pressure sensor.

## 2. Materials and Methods

### 2.1. Fabrication of SiNWs

Arrays of interconnected SiNWs were fabricated by silver (Ag) MACE in a three-step process, from p-type, single-side polished 525 μm-thick Si wafers, with resistivity, ρ, of 10 Ω cm–20 Ω cm. The nanowire patterns were made on areas of about 1 cm^2^ on the polished side of the wafers. The sequence of steps used to prepare the SiNW areas is as follows:1.Deposition of metal catalyst: Ag nanoparticles were deposited on the surface of the Si wafers by immersing the wafers in a solution of 3 M HF and 1.5 M AgNO_3_ for 60 s.2.Wire etching: The samples were etched by immersing them in HF:H_2_O_2_ (5M:0.4M) solution to obtain vertically aligned SiNWs.3.Removal of residual Ag nanoparticles: Samples were immersed in 20% *w/v* HNO_3_ to remove residual silver particles. A more detailed description can be found in Refs. [28,29].

Subsequently, 150 nm-thick aluminum electrodes were deposited on the samples by electron beam evaporation (Polyteknik Cryofox Explorer 600 LT). For the uniaxial measurements, the electrodes were deposited on the top and backside of the samples, while for the isostatic pressure measurements, the electrodes were made co-planar.

Four sets of interconnected SiNWs samples, denoted as follows, were prepared by varying the etching time: A ( 1 min), B ( 3 min), C ( 5 min), and D ( 7 min) were made for uniaxial pressure application, and sample E ( 40 min) was made for isostatic pressure testing.

Additionally, periodic SiNWs, such as sample F, were made as described in the Appendix A. Table 1 shows the list of the samples with corresponding etching times and length.

A scanning electron microscope (SEM, Zeiss Supra 35) was used to characterize the SiNW’s geometry. Top and cross-sectional SEM images were used to estimate the average diameter of the wires. The Gwyddion software for data visualization and analysis was applied to surface SEM images in order to estimate the total top area of the wires. Figure 1 shows (a,b) the cross-section of the wire array and (c) the surface area of SiNWs obtained after 7 min etching. It is worth mentioning here that the SEM analysis was carried out on the sample prior to the removal of Ag nanoparticles, which are visualized as bright spots at the base of the nanowires in the cross-section.

As can be seen in the top-view image, the wires are partly interconnected, forming a continuous rigid structure. Further, also seen from place to place are free-standing nanowires forming bundles. Such bundle formation may take place because of capillary forces acting during the drying process following the wet-etching step. In the cross-sectional image, one may observe that the length of the wires is relatively homogeneous, around 3 μm, and their typical diameter is approximately 150 nm. According to Peng et al. [5], porosity plays an essential role in the PZR response, which increases with increased porosity. The porosity is most conveniently controlled by the concentration of Ag deposition solution and etching time. In a previous study [28], it was demonstrated that the SiNWs porosity was highly affected by the concentration of the AgNO_3_ during the Ag deposition. A 1.5 mM AgNO_3_ (as used for samples A–D) provided highly porous SiNWs with maximum photoluminescence spectra intensity.

From the SEM image analysis, the wire’s cross-sectional surface coverage was estimated at roughly 28% by using the Gwyddion program, Figure 2. By counting the average number of wires on several line scans, the wire density was estimated to be $1.6\;\times \;10^{7}$ mm^2^, which corresponds to roughly $8\;\times \;10^{8}$ wires under the force meter area (7 × 7 mm).

### 2.2. Measurement Setups

For uniaxial PZR tests, the samples were clamped between a metal pin of a force meter (Mark-10, M5-012), touching the top side of the samples, and a rigid copper (Cu) plate at the backside, as shown in Figure 3. In order to improve the electrical contacts, the samples were glued to the Cu backside plate with silver paste. The force meter was mounted on a vertical rod allowing for movement on the *z*-axis (vertical) by a manually adjusted screw. Uniaxial pressure was applied to the samples by pressing the pin of the force meter into the samples’ surface with an intensity determined by the aforementioned screw. The applied force was in the range of 100 mN to 900 mN on an area of 7 × 7 mm (the cross-sectional area of the pin). Taking the wire coverage (28%) into account, this corresponds to a gauge pressure of 7 kPa–66 kPa. Figure 3 shows a sketch of the experimental setup. A Keithley 2400 SourceMeter was used to measure the resistance *R* through the sample as a function of applied pressure at a constant bias voltage of 2 V.

For PZR tests under isostatic pressure variation, a vacuum chamber was used, and the air was removed while the resistance was measured at a fixed 5 V. In this case (for sample E), the contacts were made in co-planar configuration, with separate contacts on each side of the patterned area, such that the nanowires are maximally exposed to air instead of top-and-bottom contacts as for the uniaxial measurements (performed with samples A–D). Furthermore, the wire length was increased (to 10 μm) to increase the surface area exposed to the air. The samples were mounted on a fixed sample station inside the chamber with tungsten needle tip kelvin probes as a connection.

## 3. Results

### 3.1. Electrical Response under Uniaxial Force

A maximum force of approximately 900 mN was applied to the samples using the setup shown in Figure 3. The maximum vertical force exerted on each wire thus corresponds to
$900\;\times \;10^{-3}$ N/$8\;\times \;10^{8}$ ≈ $1.1\;\times \;10^{-9}$ N/wire or $6.6\;\times \;10^{4}$ Pa (0.65 bar).

The force and the instantaneous resistance measured after applying the force to the sample with nanowires of length 3 μm (etched for 7 min) are shown in Figure 4a as a function of time. The maximum force was applied at the beginning (time zero) and then reduced step-wise while the resistance was being measured. Each force level was kept constant for roughly 50 s to confirm that the resistance value was stable with time. As clearly visible, the resistance changes significantly and inversely with pressure.

We define the relative resistance change vs. force variation for each step k=1,2,... as
(1)$$\frac{\Delta R}{R}=\frac{R_{k}-R_{1}}{R_{1}},$$
where $R_{k}$ is the average resistance obtained for the constant force step *k*. The measurements were repeated four times on each sample (A, B, C, D), each time reproducing the force steps as well as possible with the adjustable screw. A second average, this time of the relative resistance in step *k*, defined by Equation (Equation 1), obtained for the four measurements vs. the average normalized forces for each step, is presented in Figure 4b. Here, by normalized force, we mean the ratio of the force in a particular step *k* to the initial force, $F_{k}/F_{1}$. Hence, the maximum force values for each step-wise measurement are normalized to unity. Note that the resistance decreases when the force increases and that in Figure 4b, we show the negative of ΔR/R.

Increasing the length of the nanowires (by increasing the etching time) leads to higher relative resistance, although the trend appears to saturate; only a relatively small difference between the 5 min and 7 min samples has been obtained. The highest relative change (37.3%) in resistance is observed for the longest etching time (sample D) with regards to uniaxial pressure.

In Figure 5, the resistance versus applied force is shown again for sample D (blue line), now compared with the case of the bulk Si (red line). For this measurement, we replicated sample D four times, in the same conditions of a 7 min etching time, and performed the measurements on each sample replica. These measurements have been performed to test the repeatability of the sensor’s response in the same conditions as have been performed in Figure 4a. It is worth noting that the un-uniform step size observed in Figure 4a is due to the manually adjustable screw of the force meter shown in the schematic. The data displayed are average values measured on these samples (D-like), shown with the error bars. A striking difference between these two configurations (SiNWs vs. bulk Si) is observed. No measurable change in resistance is seen for the bulk Si, in very sharp contrast to the SiNWs sample.

Up to a 35% change in resistivity was observed over the pressure variation in the range of 0.1 N–0.9 N. In order to test the stability of SiNW’s PZR response and response time, we performed several sets of measurements over different samples. In this stability test, the resistance shift was measured in periods of 60 s of loading and unloading force for all the samples. All samples were loaded by 860(10) mN and then unloaded to 220(10) mN. We observe a fast response (below 0.5 s), good stability, and high sensitivity to repeated pressure changes. The results of a reproducibility test for sample D are presented in Figure 6.

Hydrogenation (exposure to hydrogen plasma) is widely used in the electronic industry to increase the mobility of charge carriers in semiconductors. It neutralizes deep and shallow defects and charged surface states [31,32]. Because the PZR effect has been attributed to surface states, we applied a hydrogenation treatment in order to explore the origin of the PZR effect in SiNWs. For hydrogen plasma treatment, we used a custom-built inductively coupled discharge setup with cylindrical geometry (290 mm long quartz tube with a diameter of 34 mm). The quartz tube was held inside a circular copper inductive coil with a diameter of 54 mm. A radio-frequency power generator CERSAR (c) (13.56 MHz) source coupled with an impedance-matching unit was utilized. For hydrogenation, a gas mixture of Ar/H_2_ (30/−70%) was used, and the throttle valves were adjusted to stabilize the gas pressure of 29 mbar. A more detailed description and schematic of the hydrogenation setup can be found elsewhere [33,34]. In Figure 7, we compare the behavior of sample D before and after the hydrogenation. The PZR effect decreases dramatically and can be attributed to the passivation of the surface states [35].

### 3.2. Electrical Response under Isostatic Pressure

Sample E (10 μm) was used for the investigation of the PZR response to isostatic pressure in a vacuum chamber. The PZR response was tested in the chamber under re-pressurizing conditions in the range of $10^{-4}$ mbar to $10^{3}$ mbar. The results are shown in Figure 8, where the resistance is plotted as a function of the pressure measured at a fixed bias voltage of 5 V. We observed a dramatic increase in the resistance by more than two orders of magnitude when pumping the air out of the vacuum chamber. We believe this is a result of combined mechanical and chemical effects: the pressure drop removes mechanical stress, and the lack of air and humidity suppresses the chemisorption. Note that the resistance values for sample E are much higher than for samples A–D because of the co-planar configuration of the contacts and also because of the larger length of the nanowires.

In the Appendix A we show, for comparison, the behavior of periodic arrays of SiNWs with increasing isostatic pressure.

## 4. Discussion

For solid materials, the inter-atomic spacing may be altered by strain. Consequently, apart from geometric changes in semiconductors, bandstructure-related details, such as bandgap or effective mass, may change, and thereby the resistivity may change as well. An applied strain changes the bandgap and the effective mass of charge carriers, which, in turn, affect the carrier concentration as well as their mobility [36]. Within a certain range of strain, this relationship is linear [22,37]. Niquet et al. [36] show that electron mobility saturates with strain. Subsequently, the PZR response saturates above a certain range of strain. When a uniaxial stress *X* is applied, the piezoresistance coefficient of the resistivity ρ in the direction of stress is defined as
(2)$$\pi_{l}=\frac{\Delta \rho }{\rho_{0}}\frac{1}{X}\;,$$
where Δρ is the stress-induced change in the resistivity and $\rho_{0}$ is the reference resistivity of the unstressed material.

In our case, a uniaxial compressive force *F* was applied to SiNWs along their length by a force meter. The stress is $X=F/A_{t}$, $A_{t}$ being the total cross-sectional area of the nanowires, which is equal to $pA_{m}$, where $A_{m}$ is the cross-sectional area of the pin pressed into the wires, and *p* is the relative cross-sectional area of the wires. Assuming the electrical resistivity of the nanowires is proportional to their resistance, Equation (Equation 2) becomes
(3)$$\pi_{l}=\frac{pA_{m}}{F}\times \frac{\Delta R}{R_{0}}\;.$$

The structure of our SiNWs array is robust and stable as the wires are partially interconnected, which provides high structural strength. Such stability and fast response of SiNWs is a desired property for many devices, such as solid-state accelerometers and bipolar transistors [38]. Further, our results are in agreement with the study of Ghosh et al. [19], in which large-diameter Si nanorod-based sensors were used for force detection.

He et al. [13], who measured PZR coefficients of single (p-doped) SiNWs, with diameters 50 nm to 300 nm, made from wafers with resistivities of 0.003 Ωcm to 10 Ω cm, found that the PZR was roughly inversely proportional to the diameter and proportional to the wafer resistivity. For a similar diameter and wafer resistivity as in our present work, ~150 nm and 10 Ω cm, respectively, the PZR coefficient $\pi_{l}$ for a single wire was of the order $10^{-7}$
${\mathrm{Pa}}^{-1}$. (According to Figure 2d of Ref. [13]). Using Equation (Equation 3), with *p* = 0.28, $A_{m}$ = 49 mm^2^, *F* = 0.8 N, and ΔR/R = 0.35, we obtain $\pi_{l}$≈ $6\times 10^{-6}$
${\mathrm{Pa}}^{-1}$, i.e., almost two orders of magnitude higher. Note that, in principle, Equation (Equation 2) does not depend on sample details, such as the number of nanowires or their configuration, which we believe plays a role in our case. Therefore, we attribute this higher value to a collective PZR effect brought about by the interaction between multiple and interconnected wires rather than the response of a single nanowire. Additionally, the pressure sensitivity of the sensor in the isostatic pressure variation given by $S=\frac{\Delta R}{R}/\Delta P$ [39], in the pressure range of $10^{-4}$ to $10^{3}\;\mathrm{mbar}$, is $9.98\times 10^{-6}\;{\mathrm{Pa}}^{-1}$. It is also seen that the highest sensitivity (highest slope in Figure 8) is found in the pressure range of roughly $10^{-2}$ and $10^{-1}$ mbar where the sensitivity measure is $8.8\times 10^{-3}\;{\mathrm{Pa}}^{-1}$.

## 5. Conclusions

In summary, large arrays of interconnected SiNWs were fabricated in a simple three-step wet chemical process and used for testing the piezoresistance effect in nanowires. The interconnected structures of the SiNWs provide a great increase in mechanical stability. A pressure change of 100 Pa could be measured with this robust device. The calculated PZR coefficient based on SiNWs array with NWs length of 3 μm as resulted after MACE etching for 7 min, sample D, was almost two orders of magnitude higher for our sensor than reported for a single SiNW. Repeated measurements for different samples fabricated with the same process demonstrated good reproducibility with less than 5% deviation in pressure sensing. The electrical resistance of SiNWs of 10 μm length increased more than two orders of magnitude when measured in a vacuum. These findings make the device based on random and interconnected SiNWs a strong candidate as a simple and inexpensive alternative to various pressure-sensing applications.

## Figures and Tables

**Figure 1 sensors-22-06340-f001:**
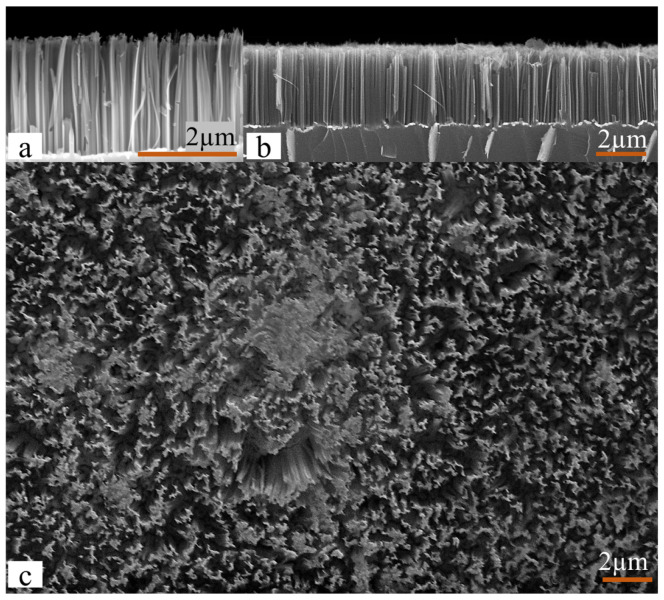
SEM image of SiNWs etched for 7 min (**a**), high-magnification cross-section of the same image (**b**), and (**c**) top-view SEM image of SiNWs etched for 7 min.

**Figure 2 sensors-22-06340-f002:**
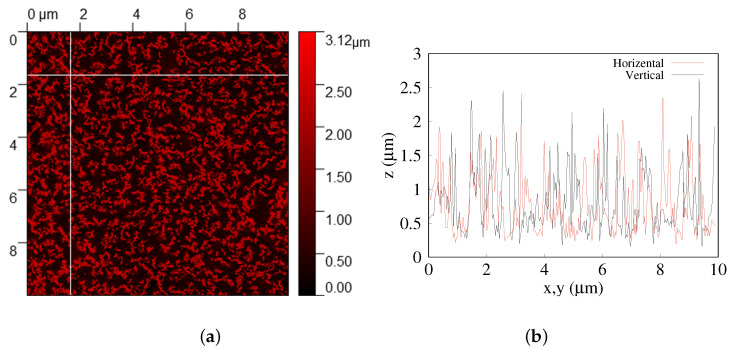
(**a**) Gwyddion analysis of top-view SEM image in (100 μm^2^). (**b**) Two Gwyddion line-scans of the figure to the left (red color—horizontal axis, black color—vertical axis).

**Figure 3 sensors-22-06340-f003:**
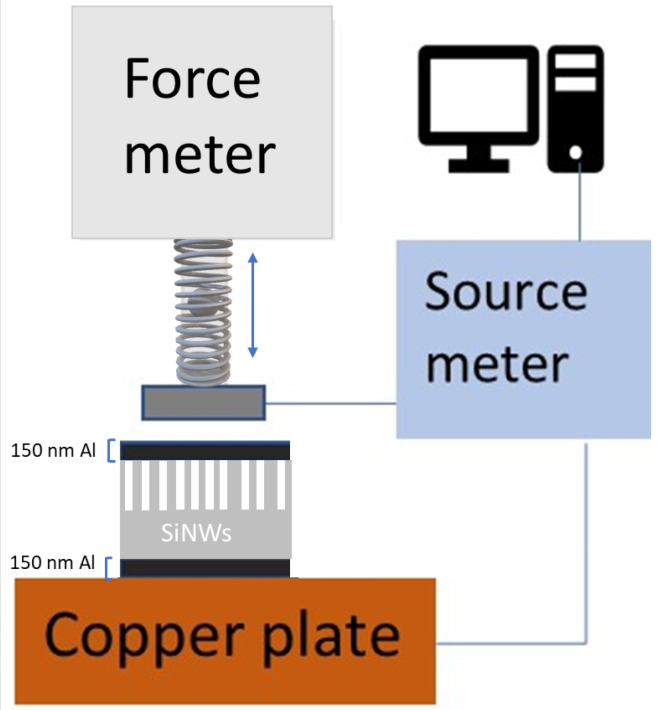
Schematic of the force meter setup for testing PZR characteristics.

**Figure 4 sensors-22-06340-f004:**
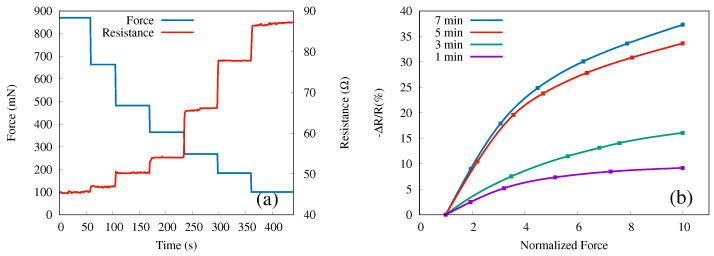
(**a**) Instantaneous resistance (right *y*-axis, red line) in response to applied force (left 4-axis, blue line) over time for sample D. (**b**) Relative resistance change versus normalized force for SiNWs samples A (purple), B (green), C (red), and D (blue).

**Figure 5 sensors-22-06340-f005:**
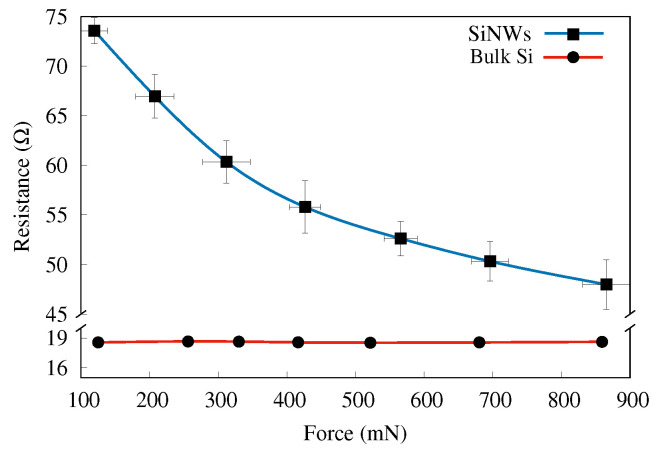
Resistance vs. applied force for the D-like samples (blue line) and bulk-Si (red line). Data for SiNWs are average values of four distinct samples; the error bars show the range in which the measurements fell.

**Figure 6 sensors-22-06340-f006:**
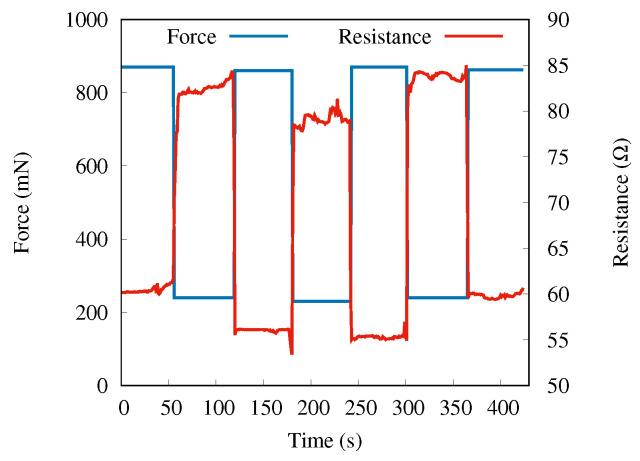
SiNWs PZR response due to repeated load–unload forces in real time for sample D.

**Figure 7 sensors-22-06340-f007:**
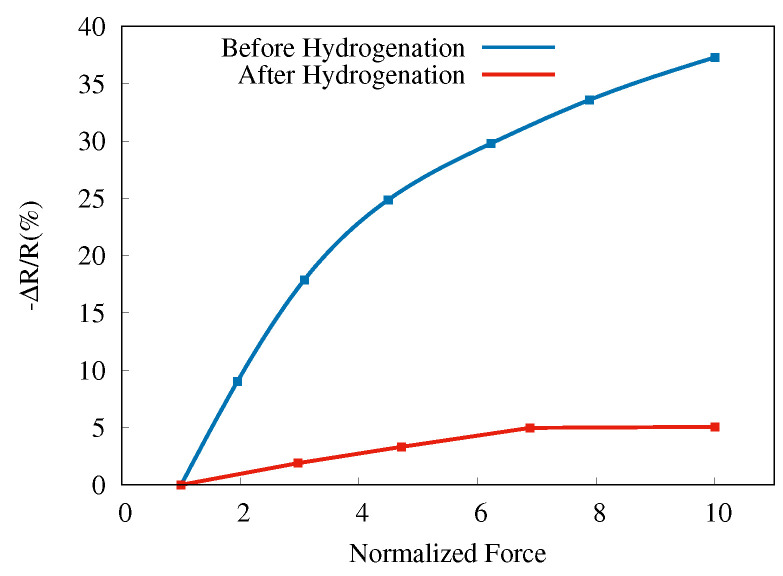
SiNWs PZR response for sample D before and after hydrogenation.

**Figure 8 sensors-22-06340-f008:**
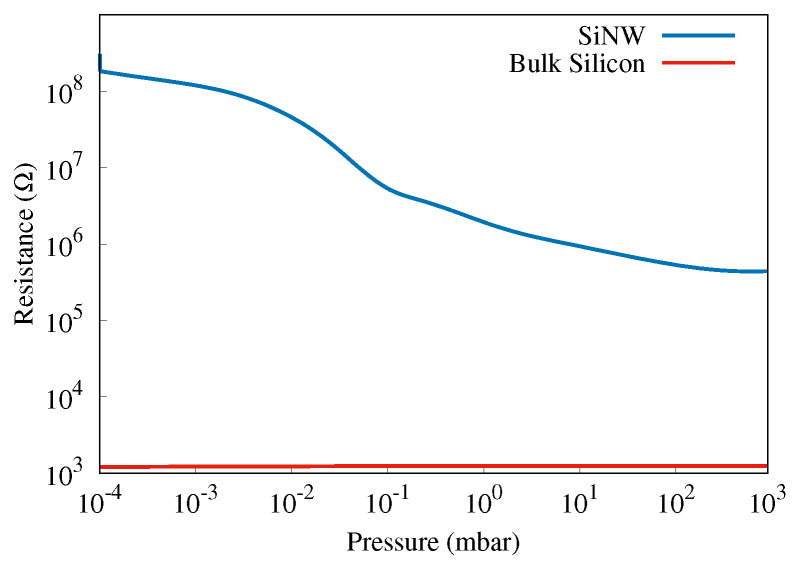
The resistance of sample E (on a log 10 scale) as a function of pressure measured in the vacuum chamber is shown with the blue curve. For comparison, in red, the resistance of a bulk silicon sample (i.e., without nanowires).

**Table 1 sensors-22-06340-t001:** List of samples with different etching times and corresponding lengths. Samples A, B, C, D, and E are random wires, and sample F is with periodic wires (shown in the Appendix A).

Sample	Etching Time (min)	SiNW Length (µm)
A	1	0.7
B	3	1.5
C	5	2.2
D	7	3
E	40	10
F	5	0.65

## Data Availability

Not applicable.

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
