# Peer review of "Piezoresistance Characterization of Silicon Nanowires in Uniaxial and Isostatic Pressure Variation"

_sensors, 2022, doi:10.3390/s22176340_

Round 1

Reviewer 1 Report

This manuscript conducted the piezoresistance characterization of silicon nanowires in uniaxial and isostatic pressure variation. The piezoresistive effect of P-type silicon nanowire obtained by metal-assisted chemical etching was studied and the piezoresistive response of silicon nanowire arrays under uniaxial and isostatic pressure was analyzed. Through experimental research and theoretical inference, the sensing resolution of silicon nanowire arrays down to about 1 mbar pressure was verified. The manuscript is interesting but still needs further revision. This reviewer has a few major comments as follows:

1.     When claiming the nanowires possess the ability to significantly increase the PZR response between line 25 to 26, it is necessary to provide the corresponding references. When referring to metal-assisted chemical etching to prepare  Si nanowires between line 38 to 40the following references are suggested to be added (Small Methods, 10.1002/smtd.202200329; International Journal of Extreme Manufacturing, 2021, 3(3): 35104; International Journal of Extreme Manufacturing, 2021, 3, 032002).

2.     In Figures 1, SEM images with larger magnifications can be added to clearly show the features of the SiNWs structure. TEM images of single NW are also suggested to be added.

3.     The "Resistance" mark in Figure 4(a) should be as far away from the curve in the figure as possible.

4.     When it is claimed that the PZR effect decreases dramatically because of passivation of the surface states (line169 to 170), please also provide the corresponding references or provide relevant explanations. When it is claimed that SiNWs array with NWs length of 3 µm was almost two orders of magnitude higher than reported for a single SiNW (line 214 to 216), please also provide the corresponding references for comparison.

5.   Although the influence of nanowire length on PZR is described in this paper through control experiments, will other parameters of nanowire affect PZR, such as nanowires density and nanowires diameter?

6.   In the discussion section, although the definition formula of the piezoresistive coefficient is given, and it is explained that the semiconductor band gap and effective mass will affect the resistivity, can a more specific mechanism be given? For example, why do the above parameters affect the PZR?

7.   The influence of nanowire length on the relative change of PZR is explored in this paper, but it seems that no direct reason is given. In Figure 4, when it is claimed that there is only a small difference between sample etched for 5 min and the 7min, and the relative resistance tends to be saturated. But it seems that more experimental data are needed to prove it, and can you explain why it is approaching saturation?

Author Response

Dear Reviewer

Please find the attachment here.

Reviewer 2 Report

Minor revision is needed before publication in Sensor:

(1) Authors are suggested calculate the pressure sensitivity. Follow and cite this article:  10.1016/j.jcis.2020.10.006.

(2) Novelty of the work should be highlighted in detail.

(3) Latest paper on piezo-resistive sensor should be cited and discuss in introduction.

Author Response

亲爱的审稿人

请在此处找到附件。

Reviewer 3 Report

The authors have presented an interesting work about the piezoresistance characterization of SiNWs. This work demonstrated the fabrication of SiNWs, and testing of SiNWs under uniaxial and isostatic pressure variations with promising outputs. However few questions can be addressed.

11. What is the repeatability of the uniaxial PZR test ( 900 to 100 mN) shown in Figure 4a?

22. If loading and unloading tests are performed, what is the hysteresis of the response of the SiNWs for uniaxial tests.?

33. The force's uniform step size was not maintained for the uniaxial test (Figure 4a). Authors need to provide the reason.

44. What is the response time of SiNWs under the uniaxial PZR test (from Figure 6)?

Author Response

(The authors gave the same response as above.)

Round 2

Reviewer 1 Report

The authors have addressed all my comments.